# The internal structure of the Brunt Ice Shelf from ice-penetrating radar analysis and implications for ice shelf fracture

Edward C. King[1], Jan De Rydt[1,2], G. Hilmar Gudmundsson[1,2]

[1]Ice Dynamics and Palaeoclimate Team, British Antarctic Survey, Cambridge, CB3 0ET, UK.

[2] Department of Geography and Earth Science, Northumbria University, Newcastle, NE1 8ST, UK.

*Correspondence to*: Edward C. King (ecki@bas.ac.uk)

**Abstract.** The rate and direction of rift propagation through ice shelves depends on both the stress field and the heterogeneity, or otherwise, of the physical properties of the ice. The Brunt Ice Shelf in Antarctica has recently developed

new rifts which are being actively monitored as they lengthen and interact with the internal structure of the ice shelf. Here we present the results of a ground-penetrating radar survey of the Brunt Ice Shelf aimed at understanding variations in the internal structure. We find that there are flow bands composed mostly of thick (c. 250m) meteoric ice interspersed with thinner (c. 150m) sections of ice shelf that have a large proportion of sea ice and sea-water-saturated firn. Therefore the ice shelf is, in essence, a series of ice tongues cemented together with ice mélange. The changes in structure are related both to

the thickness and flow speed of ice at the grounding line and to subsequent processes of firn accumulation and brine infiltration as the ice shelf flows towards the calving front. It is shown that rifts propagating through the Brunt Ice Shelf preferentially skirt the edges of blocks of meteoric ice and slow their rate of propagation when forced by the stress field to break through them, in contrast to the situation on other ice shelves where rift propagation speeds up in meteoric ice.

## 1 Introduction

Ice shelves provide an important buttressing mechanism that restrains the ice flow from ice sheet interiors towards the ocean. The disintegration of the Larsen A and B ice shelves on the east coast of the Antarctic Peninsula resulted in substantial acceleration of the glaciers that formerly flowed into the area lost (De Rydt et al., 2015, Hulbe et al., 2008, Rignot et al., 2004, Rott et al., 2011, Scambos et al., 2004, Shuman et al., 2011). These events have led to considerable current interest in

the mechanisms of ice shelf fracture and breakup and the associated consequences for the far-reaching upstream impacts of such changes (Reese et al., 2018). Work on several ice shelves has demonstrated an association between internal structure and the rate and direction of rift propagation. On the Ronne Ice Shelf Hulbe et al. (2010) noted that crack tips coincided with suture zones between ice originating from adjacent outlet glaciers through most of the advective path, with rifts propagating through the suture zones only near the ice front. On Amery Ice Shelf, Walker et al. (2015), showed rifts changing their

propagation direction several times on entering the suture zone between two ice streams. A number of studies have focussed on the Larsen C Ice Shelf since the break-up of its northern neighbour (Glasser et al., 2009, Jansen et al., 2013, Kulessa et al., 2014, McGrath et al., 2012). Here, the details of the formation of suture zones with a high proportion of marine ice in areas immediately downstream of coastal promontories has been documented (Holland et al., 2009, Jansen et al., 2013). The suture zones are composed of accreted marine ice, sea ice, fallen meteoric blocks and accumulated snow from both drift snow captured in the surface depression and direct snowfall (Jansen et al., 2013, Leonard et al., 2008, McGrath et al., 2013). Numerous rifts in the meteoric ice bands terminate against the suture zones indicating that the heterogeneous ice within the zones has higher fracture toughness, providing resistance to rift propagation through the ice shelf (Borstad et al., 2017). When the large rift in Larsen C Ice Shelf broke through the suture zone, the speed of propagation increased markedly as the rift traversed meteoric ice. Suture zones are thinner, warmer, and more heterogeneous than the meteoric ice and may have differences in water content and crystal fabric, all of which likely vary spatially throughout the zone. Which combination of these factors is most important in determining fracture toughness is still unknown. These examples from various ice shelves show that it is important to understand the internal composition and structure of ice shelves as this may impact on the rate and path of fracture. However, direct observations on the control exerted by internal structure on crack propagation are limited. Here we present for the first time direct evidence that the path of propagation is directly influenced by deep lying structures within an Antarctic ice shelf.

## 1.1 Study Area

The Brunt Ice Shelf is located on the eastern coast of the Weddell Sea (Fig. 1), and forms the southernmost portion of a complex ice shelf that incorporates the Stancomb-Wills Glacier Tongue and the Riiser-Larsen Ice Shelf to the east. The Brunt Ice Shelf flows northwest from the coast of Coats Land with a speed of >500 ma$^{-1}$ at the calving front, although there is strong temporal variability in the flow regime (Gudmundsson et al., 2017). The flow is restrained by grounding at the McDonald Ice Rumples (Fig. 1) in the northeast corner of the ice shelf (Thomas, 1973a, b, 1979).

The grounding zone between the Brunt Ice Shelf and the grounded ice in Coats Land is steep and heavily crevassed, so that the entire ice sheet breaks up into large blocks between 2500 and 6000 m long and 250 to 900 m wide. Therefore, the majority of the ice shelf within 15 to 20km of the grounding line actually comprises icebergs surrounded by sea ice and has large topographic relief (Fig. 2). Accumulation of falling and drifted snow infills the topography such that the surface of the downstream part of the Brunt Ice Shelf undulates gently giving little indication of the underlying structure. The presence of marine ice deposited at the base of the ice shelf is highly likely (Khazendar & Jenkins 2003, Khazendar et al., 2009).

In this study, we use ice-penetrating radar surveys to describe aspects of the internal structure of the Brunt Ice Shelf and interpret potential mechanisms for its development. Previous studies of the ice shelf using radio echo-sounding (Bailey and Evans, 1968; Walford, 1968) showed that echo strength from the base of the ice was highly variable and that there were regions of very high attenuation where no basal echoes where recorded. The high attenuation was attributed to the

percolation of sea water or the presence of saline ice formed from sea water. We have applied up-to-date radar sounding equipment and techniques to provide better spatial coverage and resolution than the previous efforts.

An extensive network of GPS monitoring stations has been established on the Brunt Ice Shelf as part of the infrastructure for Halley 6 Research Station (Anderson et al., 2014). Recently, two significant new rifts developed in the ice shelf (De Rydt et al., 2017), one an extension of 'Chasm 1' (Fig.1) which has remained unchanged since the 1970s, one a new rift named 'Halloween Crack' (from the date of formation on 31 October 2016). Thus the region around Halley Station is a unique place to study the impact of ice shelf structural heterogeneity on fracture propagation because dynamic changes are underway within a well-monitored environment and the station provides a logistic hub to undertake extensive ground-penetrating radar survey. The rate and direction of propagation of Halloween Crack changed when the crack tip entered a region of thicker ice shelf thought to comprise closely-spaced blocks of meteoric ice (De Rydt et al., 2017). In this paper, we use radar results to determine the degree of heterogeneity in ice shelf internal structures and discuss how this came to be.

In Section 2 the regional surface and basal topographic data is reviewed. In Section 3 the radar data acquisition and processing is described. Section 4 describes and interprets the radar profiles. In Section 5, we will discuss the radar profiling results in terms of the internal structure of the ice shelf and the implications for fracture propagation are presented in Section 6.

## 2 Observations

### 2.1 Surface topography

The surface topography (Fig. 2) is based on a number of 3 m resolution WorldView DEM tiles acquired between 2012 and 2014. To stich tiles together, one tile, which included the buildings of Halley 6 station was designated the anchor tile, adjacent tiles were shifted manually in x, y and z coordinates (without rotation) to match identifiable points between pairs, particularly in near-static areas around the McDonald Ice Rumbles and on the grounded ice sheet. There are several distinctive topographic regions as follows.

- An inner region within 20 km of the grounding line. Here there are large areas where the snow surface is between 1 and 10 m above sea level (blue colours in Fig.2).

- Curvilinear bands where the average elevation is 35 m above sea level (orange colours in Fig. 2). These bands can be traced from the grounding line to the calving front. They have a distinctive appearance created by undulations which is reminiscent of close-spaced railway sleepers. For the purposes of description, we term them 'railway tracks'. The spacing of the 'sleepers' is around 1.5 km (profile A-A', Fig. 2).

- Intervening bands where the average elevation of the ice shelf is low, in particular in the first 20 km from the grounding line large areas are within 2 m of sea level with scattered highs rising to 35-40 m. With distance from the grounding line, the broad troughs become filled in while the peaks decline in elevation from 35-40 m to 25-30 m

(Profile B-B') in Fig. 2. High points in these bands are more widely spaced and irregular than in the 'railway tracks'.

- Steep-sided, flat bottomed rifts with walls around 30 m high. These are locally known as chasms (Chasm 1 and Chasm 2, Fig. 2). The chasms extend 30 km into the ice shelf from the south.

## 2.2 Sub-glacial topography

The BEDMAP 2 database (Fretwell et al., 2013) was used to map the subglacial topography beneath the grounded ice in Coats Land (Fig. 2). Within the mapped area there are two 10 km wide troughs with an intervening ridge that are oriented approximately orthogonal to the grounding line. Ice flow speed derived from InSAR data (Rignot et al., 2017) shows that within the troughs flow speed peaks at 105 ma$^{-1}$ and 100 ma$^{-1}$, whereas ice flowing off the ridges has a flow speed of between 15 and 35 ma$^{-1}$ (Fig. 3).

## 3 Radar acquisition and processing

The data were collected between December 2015 and February 2016 using a commercial ground-penetrating radar system (Sensors and Software PulseEKKO PRO) operating at a centre frequency of 50 MHz. The system was mounted on a sledge, which was towed behind a snow mobile travelling at 15 km/hr, with traces recorded continuously. Positioning information was recorded using a dual-frequency GPS receiver and the satellite range information was processed through the Canadian Geodetic Service Precise Point Positioning service in kinematic mode. This methodology provided radar profiles with positions accurate to around 0.5 m. The radar data were processed using ReflexW software by applying a bandpass filter (with corner frequencies 10/20/60/120 MHz), a spherical spreading correction, a horizontal filter to remove the direct arrival, and time migration. A fixed wave speed of 0.168 m/ns was used for both migration and depth conversion because the construction of a detailed wave speed model for the ice shelf using multiple common mid-point determinations was beyond the scope of the survey. We therefore elected to use the fresh water ice wave speed of 0.168 m/ns throughout. We estimate that our figures for the thickness of the ice shelf where most of that thickness is made up of meteoric ice have an uncertainty of about 20%. Where there is significant thickness of pure firn (i.e. not infiltrated with sea water), the use of a uniform wave speed underestimates the overall ice shelf thickness by approximately 5%.

## 4 Radar data

### 4.1 Data description

In this section, we will first describe the reflection character of some example radar profiles taken from a large data set. We will then interpret the different reflection facies observed in terms of their glaciological origins.

Portions of two flow-line radar profiles 1800 m long are presented in Figure 4. The locations are shown in Fig. 2. Line 04 was acquired in one of the 'railway track' bands of slightly higher topography. Line 62 was acquired in the adjacent lower topography region to the west of Line 04.

Line 04 (Fig. 4a,b) has a near-surface radar facies of undulating, layered, continuous reflections between 13 and 60 m thick. This facies is colour-coded yellow in the figures. The layered facies lies above a second facies with irregular, scattered

reflections of highly variable amplitude (colour-code blue). The deepest sections of the interface between the two facies are characterized by very high amplitude, laterally continuous reflections (orange). The remainder of the profile is largely reflection-free until a band of curvilinear reflections at between 240 and 270 m below surface. The events marked as multiples are reflections that arise from internal reverberation between strong reflectors.

Line 62 (Fig. 4c,d) also has an upper radar facies of undulating, layered, continuous reflections, this time between 13 and 40

m thick. There are three regions with irregular, scattered reflections of variable amplitude forming prominent highs in the section. Between these highs there is a near-horizontal reflection with very high amplitude at approximately 40 m below surface (the surface is at 22 m above sea level). This reflection has a prominent multiple. Below the three highs, there are weak, scattered reflections then, at around 150 m depth, a set of high-amplitude, curvilinear reflections.

Figure 5 shows another example and more detailed view of the radar reflection facies described above. Line 05 (for location

see Fig. 2) is located in one of the lower topography bands in the ice shelf but adjacent to some isolated ridges. The upper radar facies of undulating, layered, continuous reflections is between 21 and 50 m thick. The second facies of irregular, scattered reflections of variable amplitude forms a single high between 450 and 680 m along the profile. Elsewhere, the base of the layered facies is a very high amplitude reflection which is conformable to the layers above in some places and is flat-lying between 170 and 330 m along the profile.

### 4.2 Interpretation

The undulating, layered, continuous reflection facies is interpreted as firn deposited in-situ on the ice shelf. The reflections are isochrones, therefore the spacing between them gives an indication of the local relative accumulation rate e.g. (Vaughan et al., 2004, Vaughan et al., 1999). The radar system does not have sufficient vertical resolution to image individual annual layers, the reflections observed are the result of the convolution of returns from many, more finely-spaced, reflectors.

The radar facies with irregular, scattered reflections of variable amplitude is interpreted as the returns from blocks of meteoric ice embedded in the ice shelf (this facies is coloured blue in Figs. 4 and 5). Figure 6 shows a visual satellite image of the region around the grounding line (the location is shown as a black rectangle on Fig. 2). It is evident that the ice

flowing over the grounding line is completely riven by crevassing and that the ice shelf at this point comprises more or less closely packed icebergs held together by sea ice (Fig. 6b). The curvilinear reflections at depth on Lines 04 and 62 (Fig. 4) are interpreted as arising from the ice/water interface at the base of the embedded icebergs. The undulations of the firn reflections in the upper radar facies (yellow colouring) indicate differential accumulation between and over the embedded icebergs (Figs. 4 and 5). It is also possible that undulations in the firn may, in part, be due to horizontal shortening induced by historic changes in flow speed of the ice shelf (Gudmundsson et al., 2017).

The very high amplitude reflection (orange colour in figures) that is near-horizontal on Figure 4c and in parts cross-cutting and parts conformable with the firn layering in Figure 5 is interpreted as a brine infiltration front. The reflection cross-cuts the isochronal reflections, suggesting a later, non-stratigraphic origin. There is strong attenuation beneath the reflector which, together with the very high amplitude, suggests a large contrast in electrical conductivity. Liquid brine was found in a hole drilled in 'thin ice shelf' to the west of the MacDonald Ice Rumples (Thomas 1973). The brine was found at a depth 1.5m below sea level. Its temperature was -10° C, suggesting a salinity of 125 ‰. Immediately below the brine was bubbly impermeable ice. Brine infiltration has been observed on radar profiles of the McMurdo Ice Shelf (Grima et al., 2016, Kovacs & Gow 1975, Morse & Waddington 1994) as well as the Wordie Ice Shelf, the Larsen Ice Shelf and Wilkins Sound (Smith & Evans 1972).

The brine reflection observed in Figs. 4 and 5 is considerably deeper than sea level at those locations. Figure 7 shows an example of a radar profile that approaches one of the large rifts in the ice shelf. On this profile there are very high amplitude near-horizontal reflections at two levels. The upper level is close to sea level while the other is 13m lower. We interpret this profile to indicate that there has been recent brine infiltration horizontally from the rift where sea water can be observed in cracks in the sea ice flooring the chasm. We suggest that the other reflection arises from an older brine infiltration event, which may have a different mechanism which will be discussed in the next section.

## 5 Discussion

We have established that the 'railway track' bands of higher elevation originated at the grounding line in locations where there are troughs in the bed topography. Conversely the bands of lower topography originated at ridges in the bed topography beneath Coats Land. Figure 8 summarizes the situation in cross-section. Ice flowed from the troughs in bedrock at a higher rate than off the ridges, providing a steady stream of large, thick blocks of meteoric ice that formed a closely-packed flow band within the ice shelf. Ice that flowed off the ridges in the bed topography was thinner and supplied to the ice shelf at a slower rate. As a result, the meteoric ice blocks were both thinner and more spaced out, resulting in isolated icebergs surrounded by large areas of sea ice (Fig. 2, topographic profile B-B'; Fig. 3c; Fig. 6b).

The majority of Antarctic ice shelves are formed by the coalescence of glaciers or ice streams that flow across the grounding line in structurally coherent bodies, perhaps with some surface or bottom crevassing, but otherwise intact. The Brunt Ice Shelf is one of a class of ice shelves (other examples are Thwaites Glacier Tongue, the western sector of Cook Ice Shelf and

the ice shelf lying off the Leopold and Astrid Coast) comprising ice which retains no structural integrity when flowing across the grounding line, and as a result, the blocks of meteoric ice are cemented together by sea ice and drift snow to form the ice shelf.

The process which cements together the ice shelf involves the gaps between meteoric ice blocks being filled first by sea ice and then by drift snow which consolidates into firn (Fig. 9 a&b). The isostatic loading by snow accumulation pushes the sea ice downwards until it lies below sea level. At this point, the process of firn consolidation has not advanced to pore close-off and the firn is still porous and permeable, allowing sea water to soak the firn. Freezing cycles then concentrate the salt to leave a brine horizon (Fig. 9c). In Figure 5 the topography on the high-amplitude reflector provides evidence that the brine infiltration may have occurred by sinking of the firn rather than by horizontal spreading of salty water. Warm periods in the summer can produce melt horizons within the firn, creating an impermeable layer. Our hypothesis is that the soaking of the firn from below as it sank to ocean level may have been blocked by the impermeable layer in the region between 50 and 150 m along Line 05 (Fig. 5). This would explain why the brine infiltration reflector is a syncline conformable with the firn layering in this section of profile. The same process may have occurred between 670 and 800 m along the profile, although it is also possible that enhanced accumulation in this region created or deepened the syncline there after the brine horizon had become frozen into place. Another possible explanation of the syncline in the brine reflector between 670 and 800m is horizontal shortening induced by variations in the flow speed of the ice shelf.

It is not clear what controls the termination of the process of brine infiltration and locks the high-amplitude reflector in place within the ice shelf so that it descends below sea level as further accumulation takes place. It is likely that a balance between pore space reduction by compaction of the firn and freezing of sea water in the gaps between ice crystals eventually reduces the permeability of the firn to nothing at a depth shallower that the dry firn pore close-off would be. Another factor may be the formation of marine ice below these sections of developing ice shelf in similar fashion to the formation of marine ice in rifts described by Khazendar and Jenkins (2003). While marine ice is porous and permeable when first formed, it thickens and consolidates over time and could eventually close off access for sea water to the lower sections of firn.

The lower elevation bands between the 'railway tracks' in the Brunt Ice Shelf have many of the characteristics of ice mélange, that is they combine sea ice, marine ice, firn and scattered meteoric ice blocks. Ice mélange has been identified as a prominent feature of the Brunt Ice Shelf/Stancomb Wills Ice Tongue system (Khazendar et al., 2009) where large areas were identified on either side of the Stancomb-Wills Ice Tongue cementing together large tabular icebergs. Thought of in this way, the Brunt Ice Shelf is a series of ice tongues cemented together by ice mélange to create a single mass with highly heterogeneous properties.

This heterogeneity of structure has a number of potential implications. For example, the icebergs of meteoric ice may have a different bulk density compared to the adjacent mixture of firn, sea ice and marine ice in the mélange. If the bulk density was significantly different, it would affect the thickness-from-freeboard calculations that are carried out over ice shelves where there are no ground-truth thickness measurements (Griggs and Bamber 2011). Khazender et al., (2009) estimated the

temperature of ice mélange at between -11° C and -7° C and that of meteoric ice at between -21° and -15° C. This has implications for melt rate calculations and the different types will produce meltwater of different salinity.

The propagation of a large rift through the Larsen C Ice Shelf was shown by Borstad et al., (2017) to be faster through meteoric ice than through suture zones, implying greater fracture toughness in the suture zones. It is not known whether the mechanical properties of the ice in suture zones constrained between large homogeneous bodies of meteoric ice such as those in the Larsen C Ice Shelf are similar to the mechanical properties of the ice mélange in the Brunt Ice Shelf. There are strong similarities in the way the material forms but post-formation history may be important in the development of the mechanical properties. For example suture zones can experience high shear strain (Jansen et al., 2013) whereas the ice mélange areas of the Brunt Ice Shelf probably do not.

## 6 Implications for fracture propagation

The heterogeneous structure of the ice shelf influences the rate and direction of the propagation of fractures. Over about half the area of the ice shelf, the location of the meteoric icebergs can be identified from the surface topography (Fig. 2). Where firn accumulation has buried the icebergs completely in the outer part of the ice shelf, the location of the meteoric ice can be mapped using SAR satellite imagery. The top panel in Figure 10 shows a radargram of the upper 90m of ice shelf in the central railway track, following the red line in Figure 11 from its most western point towards the grounding line. The black line in Figure 10 marks the interface between radar facies 1 (firn) and facies 2 and 3 (brine and meteoric ice). The bottom panel shows the backscatter amplitude along the same section from a Sentinel 1A radar image acquired on 29 October 2017. A Gaussian filter with a radius of 40m was applied to suppress small-scale noise. There is a very high correlation (r = 0.607, p=0) between peaks in the backscatter amplitude and the zones of thin firn overlying the crests of the icebergs. This correlation is not a coincidence, as the C-band sensor on Sentinel1-A is known to penetrate the surface of the ice (Bingham & Drinkwater 2000), and it is therefore capable of picking up the spatial variability in the structure of the ice near the surface. However, the details of this mechanism remain subject to future study, and here we merely note its existence. Using radar backscatter from Sentinel-1A as a proxy, the spatial distribution of meteoric ice can be mapped across the entire ice shelf, as illustrated in Figure 11.

The complete map of ice shelf heterogeneity provides a unique opportunity to interpret observed changes in the trajectory and propagation speed of two major rifts (Chasm 1 and Halloween Crack) in the ice shelf, and to relate these changes to the internal structure of the ice shelf. To highlight the different ways in which the rifts interact with the internal structure, we focus on two small regions outlined by the black boxes (A and B) in Figure 11. Region A (Fig. 11) covers the tip of Chasm 1 in November 2017; a detailed view is shown in Figure 12. The overall direction of propagation of Chasm 1 is dictated by the large-scale stress field in the ice shelf, with a trajectory that is perpendicular to the direction of maximum tensile stress (De Rydt et al., 2017, Gudmundsson et al., 2017). However, Figure 12 shows that at smaller length scales, the exact trajectory is dictated by the location and shape of the meteoric icebergs, and Chasm 1 follows pre-existing lines of weakness along the

edges of the icebergs, in particular within the 'railway track'. Only at one instance, Chasm 1 propagated through an area of meteoric ice (red circle in Figure 12), which coincided with a period of significantly slower propagation rates (De Rydt et al., 2017).

Region B (Fig. 11) covers part of the Halloween Crack where the rift propagated through the northern railroad track; a detailed view is shown in Figure 13. The overall direction of propagation is again dictated by the stress distribution in the ice shelf, which forces the Halloween Crack to cross an area with a high concentration of meteoric ice, where the edges of the iceberg are misaligned with the overall direction of propagation (red ellipse in Figure 13). This has resulted in a more complex propagation behaviour, where the rift followed edges of the icebergs for short periods, and broke through the icebergs in places of weakness, such as discontinuities in the 'railway sleepers'. For the remainder of the trajectory shown in Figure 13, the rift, while generally following a path dictated by the stress field, in detail weaves its way around scattered icebergs within the ice shelf, avoiding fracturing through areas of meteoric ice. Thus, fracture through the Brunt Ice Shelf progresses at contrasting rates to rifts through other documented ice shelves. Elsewhere rift extension rates increase in meteoric ice and decrease in suture zones comprising ice mélange, whereas recent Brunt rifts slow in meteoric ice and speed through ice mélange. This suggests that there is a difference in the physical properties of ice that was formed in similar ways but which had different subsequent histories e.g. amount of shear strain, during advection though the ice shelf.

## 7 Conclusions

Unusually the Brunt Ice Shelf is composed of alternating bands of ice of two types which have different origins and compositions. The first type is identifiable on visual satellite imagery by a 'railway track' appearance and comprises large blocks of meteoric ice originating in the ice sheet in Coats Land which are cemented together by thin strips of sea ice and firn. This type is mostly thick freshwater ice. The second type has a random appearance on satellite images and comprises an ice mélange of scattered, relatively thin blocks of meteoric ice from the continent separated by large areas of sea ice (probably underlain by marine ice) and firn that has been soaked by sea water and refrozen.

While a fracture propagating through the Larsen C Ice Shelf sped up when traversing through meteoric ice and slowed in suture zones, fractures observed in the Brunt Ice Shelf slowed when going through meteoric ice blocks and often routed through the ice mélange in the gaps between the meteoric ice. This contrast in styles of fracture propagation in different ice shelves needs to be better understood so as to improve predictive modelling of ice shelf stability.

## Competing interests

The authors declare no competing interests.

## Acknowledgements

The authors are grateful for the support provided by the station personnel of Halley 6 Research Station. This work was part of the British Antarctic Survey programme 'Polar Science for Planet Earth' funded by the Natural Environment Research Council, UK. The authors thank A. Booth, D. McGrath and an anonymous referee for their useful comments for improving
the paper.

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

**Figure captions**

Figure 1: The Brunt Ice Shelf lies off Coats Land on the east side of the Weddell Sea, Antarctica (inset). The surface of the East Antarctic ice sheet slopes steeply down to the grounding line marked by a white line, elevation contours are at 100 m intervals. The ice shelf is partially grounded at the McDonald Ice Rumples. After several decades of sustained ice shelf growth, two rifts have developed in the past two years, marked by red lines. The first was an extension of a dormant rift known locally as Chasm 1, the second appeared on 31st October 2016 and is known as Halloween Crack. Halley Research Station was re-located to the position marked 'Halley 6a' in January 2017. Projection is Antarctic Polar Stereographic (WGS84, EPSG: 3031)

Figure 2: Surface elevation of the Brunt Ice Shelf and the bedrock elevation beneath grounded ice in Coats Land. Flow lines downstream from troughs in the bedrock have higher elevation throughout the ice shelf (orange colours). Near to the grounding line these elevated bands comprise closely-spaced ridges resembling railway sleepers, hence we term the bands 'Railway Tracks' for descriptive purposes. White lines show locations of elevation profiles extracted from the digital elevation model and shown below the map. Line locations in continuous black mark sections of radar profiles shown in Figs. 4 and 5 (dashed black lines over profiles show full extent of profiles). Pink line is location of flow speed profile shown in Fig. 3. Box is the extent of the satellite image shown in Fig. 6.

Figure 3: a) Flow speed profile across a gateway 5km upstream of the grounding line (marked in Fig. 2). b) Surface and bed elevations for the gateway.

Figure 4: Ice shelf cross-sections acquired with a 50 MHz ground-penetrating radar system. Locations are marked in Fig. 2. Radar facies descriptions and interpretations are given in the text a) Line 04 lies along one of the 'Railway Tracks'. Three radar facies are identified (inset). b) Interpretation of Line 04. Radar facies 1 is interpreted as firn accumulated by snow fall and drift on the ice shelf. Radar facies 2 is interpreted as a brine horizon. Radar facies 3 is interpreted as blocks of meteoric ice that originate as icebergs calved off the inland ice sheet at the grounding line, the blocks are up to 200 m thick. c) and d) Radar cross-section from Line 62 in the thinner ice shelf area between two of the 'Railway Tracks'. The icebergs are significantly thinner and more widely spaced.

Figure 5: Close-up of a section of radar profile (Line 05, location on Fig. 2) that illustrates different structure on the brine reflection. In some places the reflection is horizontal and cross-cuts reflections in the firn layer. Elsewhere the reflection is conformable with the isochronal reflections in the firn layer.

Figure 6: Landsat image of the region around the grounding line (location Fig. 2). Intense crevassing occurs 3-4 km upstream of the grounding line. The ice retains no structural integrity on crossing the grounding line, the ice shelf is composed entirely of separated blocks held together by sea ice. Where the ice flows from a subglacial trough the blocks

remain closely-packed with narrow channels of sea ice between them, but where the ice flows off a subglacial ridge, the blocks are widely-spaced with extensive areas of sea ice between.

Figure 7: Westernmost section of radar profile from Line 62 (Fig. 2) near Chasm 1. High amplitude reflections interpreted as brine infiltration horizons are observed at 26 m and 39 m below the surface of the ice shelf. The depth of the upper reflection

coincides with sea level in the adjacent rift suggesting horizontal migration of sea water through porous and permeable firn recently exposed to the ocean. The lower reflection is interpreted as sea water infiltration that became frozen in place some distance upstream.

Figure 8: Cartoon to illustrate the origin of the two different structures within the Brunt Ice Shelf. A) Ice flowing out of a

subglacial trough at around $100ma^{-1}$ breaks up into closely-packed icebergs separated by narrow channels in which sea ice forms. B) Where thinner ice flows over the grounding line at a slower rate the supply of ice is insufficient to match the flow speed of the ice shelf (which is driven by the faster ice coming out of the troughs), so the structure comprises thin, widely-spaced icebergs separated by wide expanses of sea ice.

Figure 9: Cartoon to illustrate the process by which sea ice between icebergs becomes loaded by firn accumulation and driven below sea level by hydrostatic adjustment, allowing sea water to soak into the porous and permeable firn from below.

Figure 10: Top panel: The upper 90 m of a radargram along the railway track near Halley 6 Research Station compared to (bottom panel) the Sentinel-1 backscatter amplitude along the same line.

Figure 11: Map showing radar backscatter proxy for the presence of meteoric ice. Over most of the ice shelf, black indicates the presence of meteoric ice and white areas correspond to infill by sea ice and firn. Boxes indicate the location of detailed views in Figures 12 and 13. Red line is the location of the profile in Fig. 10.

Figure 4: Detailed view of the tip of the Chasm 1 Crack (blue curve). The background image distinguishes areas with meteoric ice (dark) from areas with sea-ice overlain by firn. When the crack went through one of the blocks of meteoric ice rather than around it (red circle), the rate of propagation slowed.

Figure 13: Detailed view of part of the Halloween Crack (blue curve). Crack propagation slowed when the crack crossed one of the 'railway tracks' at a high angle to the meteoric ice blocks. Rift extension speed then increased in the ice mélange beyond.

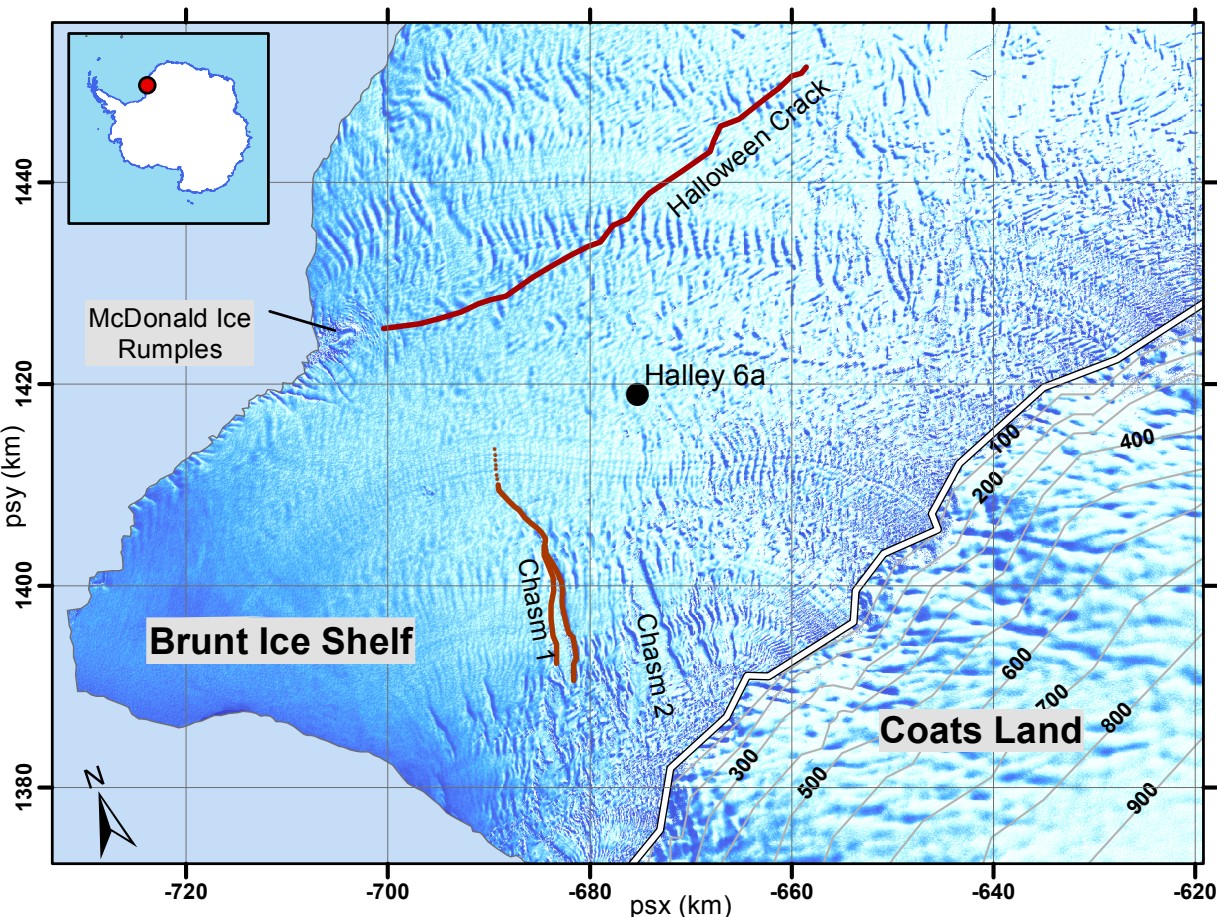

Figure 1: The Brunt Ice Shelf lies off Coats Land on the east side of the Weddell Sea, Antarctica (inset). The surface of the East Antarctic ice sheet slopes steeply down to the grounding line marked by a white line, elevation contours are at 100 m intervals. The ice shelf is partially grounded at the McDonald Ice Rumples. After several decades of sustained ice shelf growth, two rifts have developed in the past two years, marked by red lines. The first was an extension of a dormant rift known locally as Chasm 1, the second appeared on 31st October 2016 and is known as Halloween Crack. Halley Research Station was re-located to the position marked 'Halley 6a' in January 2017. Projection is Antarctic Polar Stereographic (WGS84, EPSG: 3031).

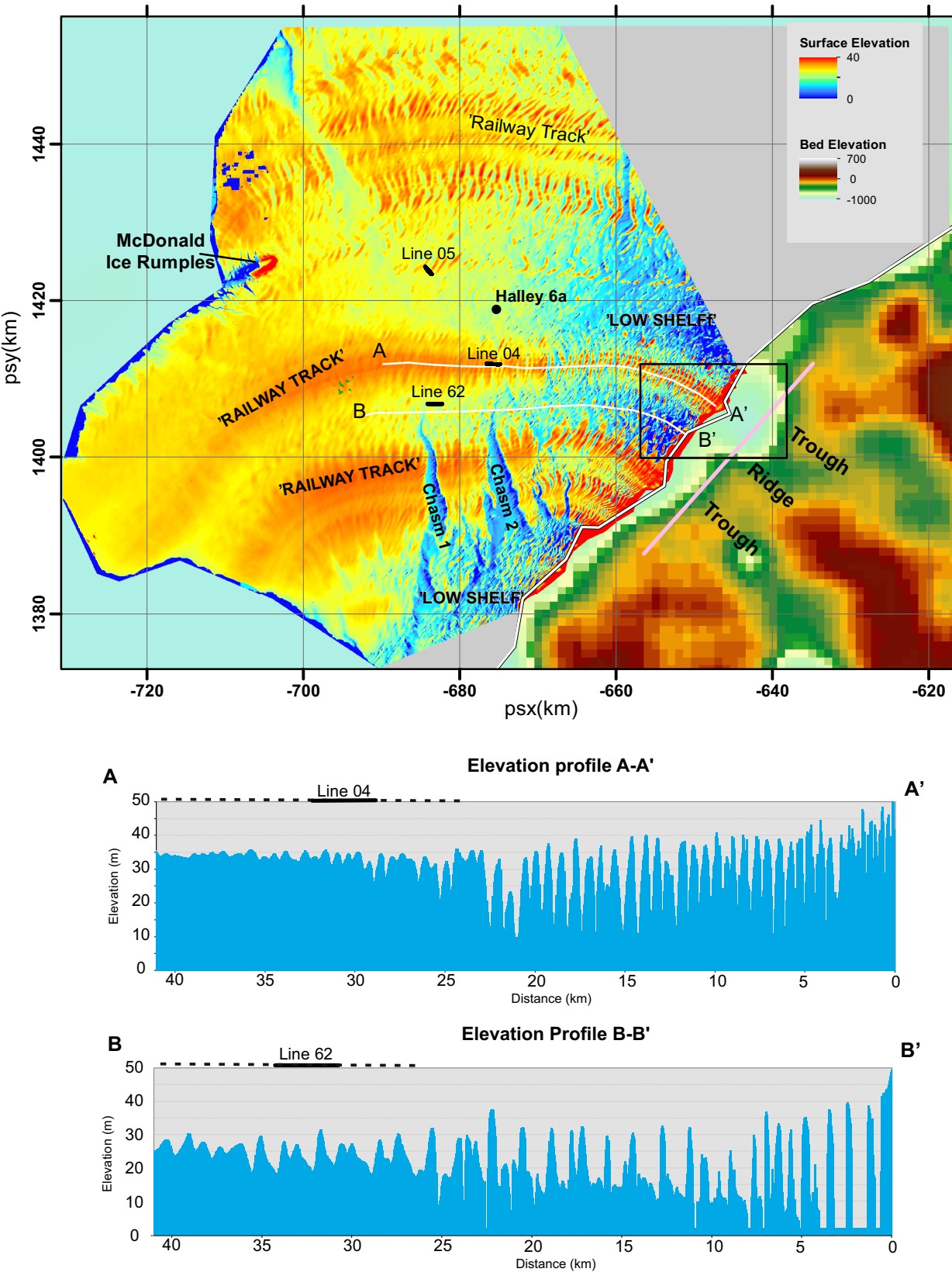

Figure 2: Surface elevation of the Brunt Ice Shelf and the bedrock elevation beneath grounded ice in Coats Land. Flow lines downstream from troughs in the bedrock have higher elevation throughout the ice shelf (orange colours). Near to the grounding line these elevated bands comprise closely-spaced ridges resembling railway sleepers, hence we term the bands 'Railway Tracks' for descriptive purposes. White lines show locations of elevation profiles extracted from the digital elevation model and shown below the map. Line locations in continuous black mark sections of radar profiles shown in Figs. 4 and 5 (dashed black lines over profiles show full extent of profiles). Pink line is location of flow speed profile shown in Fig. 3. Box is the extent of the satellite image shown in Fig. 6.

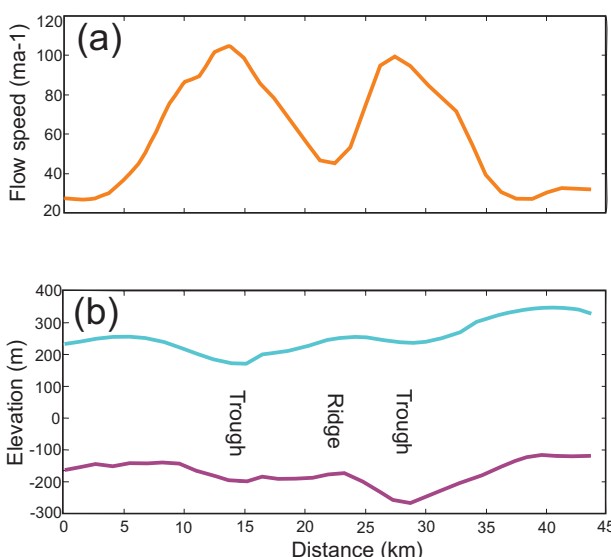

Figure 3: a) Flow speed profile across a gateway 5km upstream of the grounding line (marked in Fig. 2). b) Surface and bed elevations for the gateway.

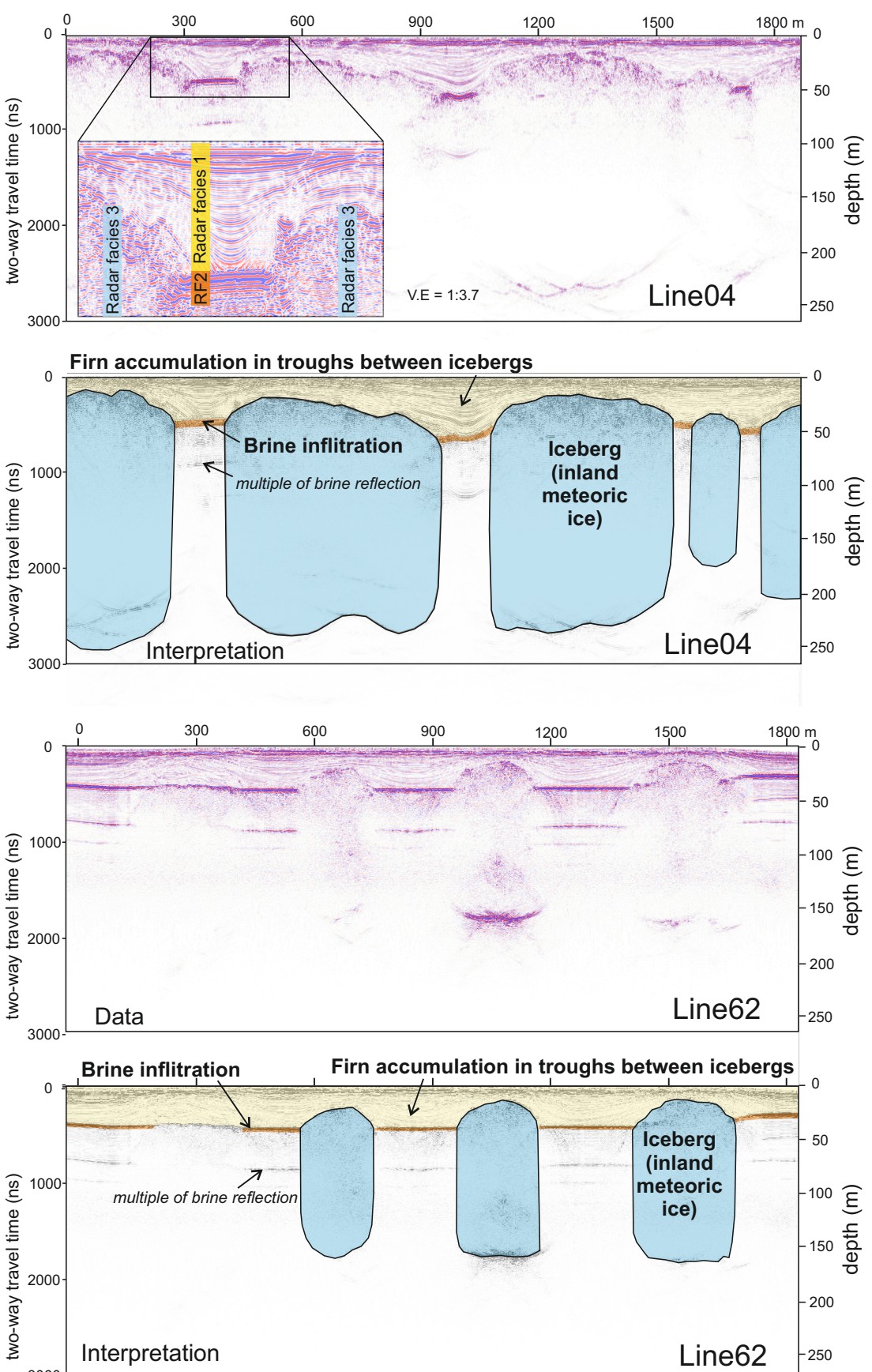

Figure 4: Ice shelf cross-sections acquired with a 50 MHz ground-penetrating radar system. Locations are marked in Fig. 2. Radar facies descriptions and interpretations are given in the text a) Line 04 lies along one of the 'Railway Tracks'. Three radar facies are identified (inset). b) Interpretation of Line 04. Radar facies 1 is interpreted as firn accumulated by snow fall and drift on the ice shelf. Radar facies 2 is interpreted as a brine horizon. Radar facies 3 is interpreted as blocks of meteoric ice that originate as ice bergs calved off the inland ice sheet at the grounding line, the blocks are up to 200 m thick. c) and d) Radar cross-section from Line 62 in the thinner ice shelf area between two of the 'Railway Tracks'. The ice bergs are significantly thinner and more widely spaced.

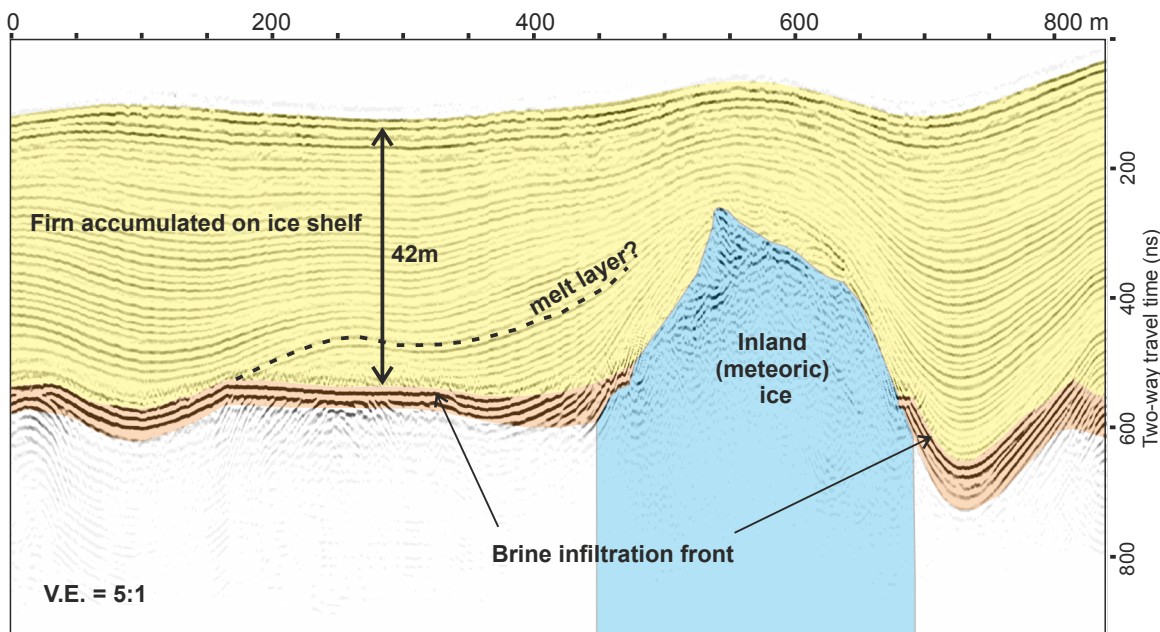

Figure 5: Close-up of a section of radar profile (Line 05, location on Fig. 2) that illustrates different structure on the brine reflection. In some places the reflection is horizontal and cross-cuts reflections in the firn layer. Elsewhere the reflection is conformable with the isochronal reflections in the firn layer.

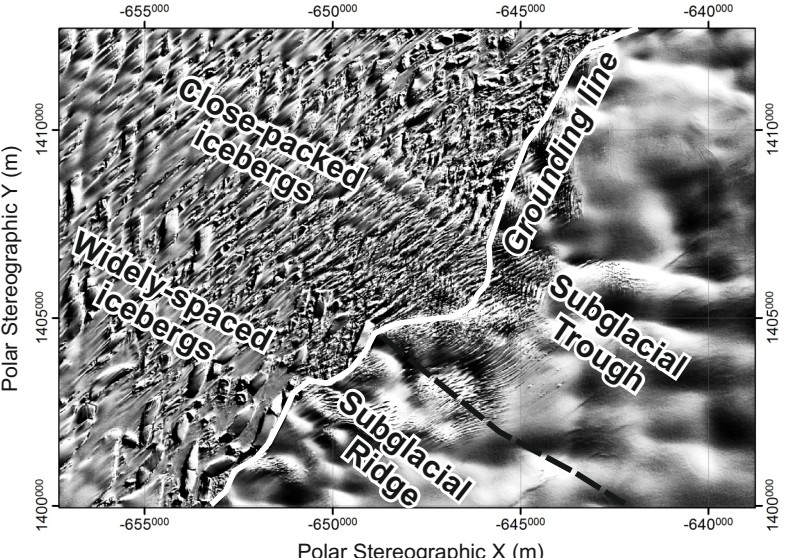

Figure 6: Landsat image of the region around the grounding line (location Fig. 2). Intense crevassing occurs 3-4 km upstream of the grounding line. The ice retains no structural integrity on crossing the grounding line, the ice shelf is composed entirely of separated blocks held together by sea ice. Where the ice flows from a subglacial trough the blocks remain closely-packed with narrow channels of sea ice between them, but where the ice flows off a subglacial ridge, the blocks are widely-spaced with extensive areas of sea ice between.

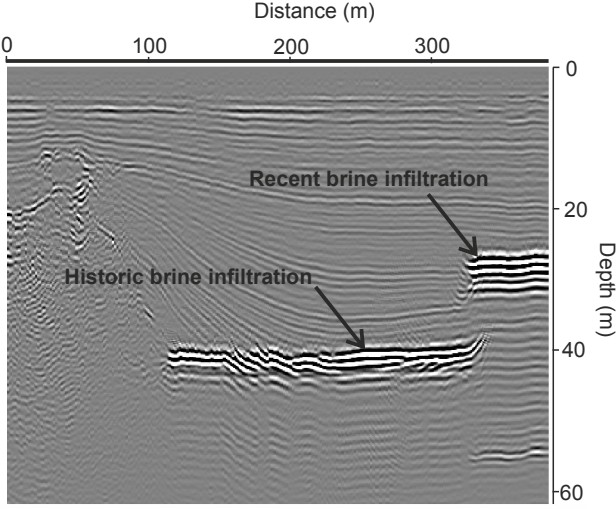

Figure 7: Westernmost section of radar profile from Line 62 (Fig. 2) near Chasm 1. High amplitude reflections interpreted as brine infiltration horizons are observed at 26 m and 39 m below the surface of the ice shelf. The depth of the upper reflection coincides with sea level in the adjacent rift suggesting horizontal migration of sea water through porous and permeable firn recently exposed to the ocean. The lower reflection is interpreted as sea water infiltration that became frozen in place some distance upstream.

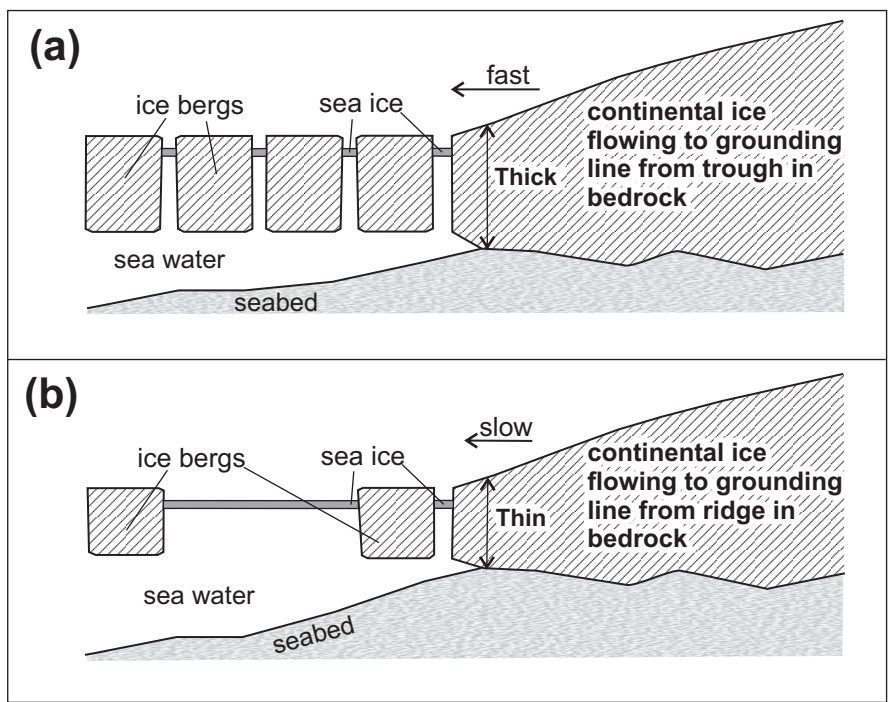

**Figure 8**: Cartoon to illustrate the origin of the two different structures within the Brunt Ice Shelf. a) Ice flowing out of a subglacial trough at around 100ma$^{-1}$ breaks up into closely-packed icebergs separated by narrow channels in which sea ice forms. b) Where thinner ice flows over the grounding line at a slower rate the supply of ice is insufficient to match the flow speed of the ice shelf (which is driven by the faster ice coming out of the troughs), so the structure comprises thin, widely-spaced icebergs separated by wide expanses of sea ice.

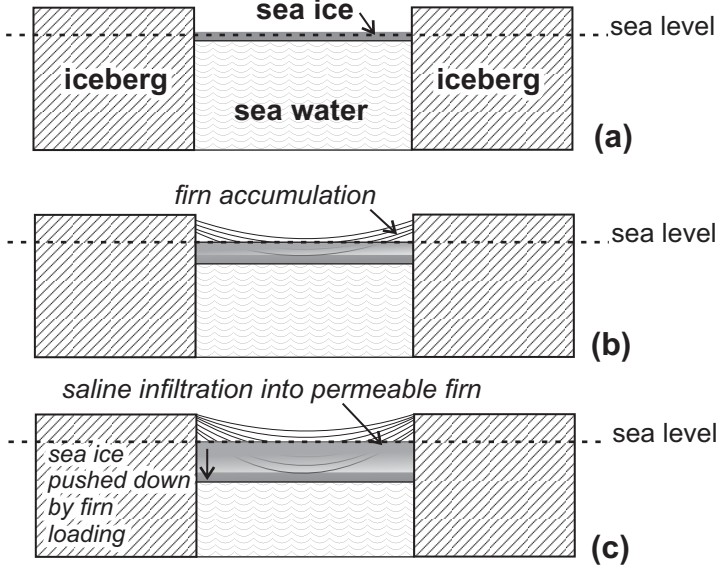

Figure 9: Cartoon to illustrate the process by which sea ice between icebergs becomes loaded by firn accumulation and driven below sea level by hydrostatic adjustment, allowing sea water to soak into the porous and permeable firn from below.

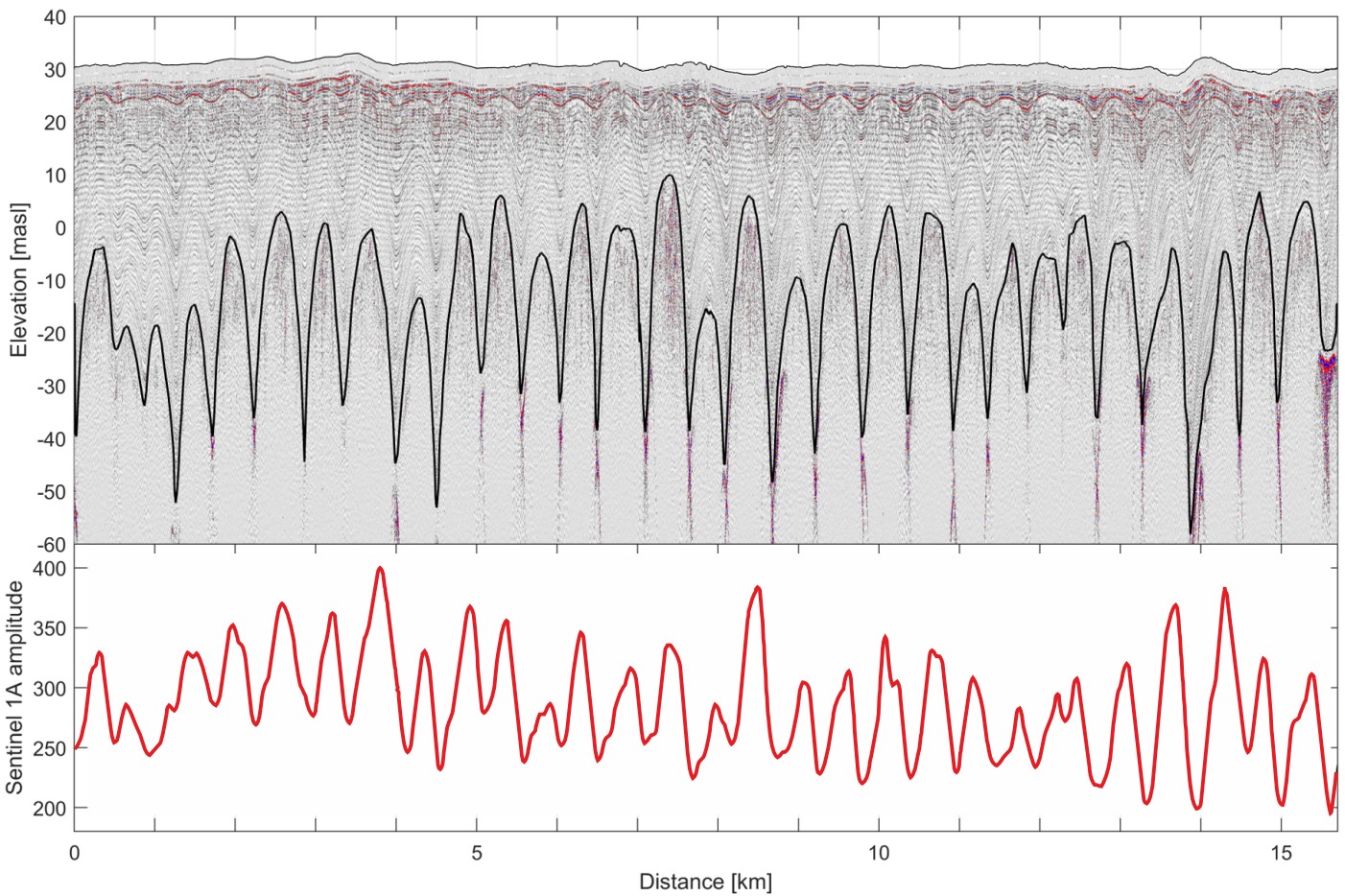

Figure 10: Top panel: The upper 90 m of a radargram along the railway track near Halley 6 Research Station compared to (bottom panel) the Sentinel-1 backscatter amplitude along the same line.

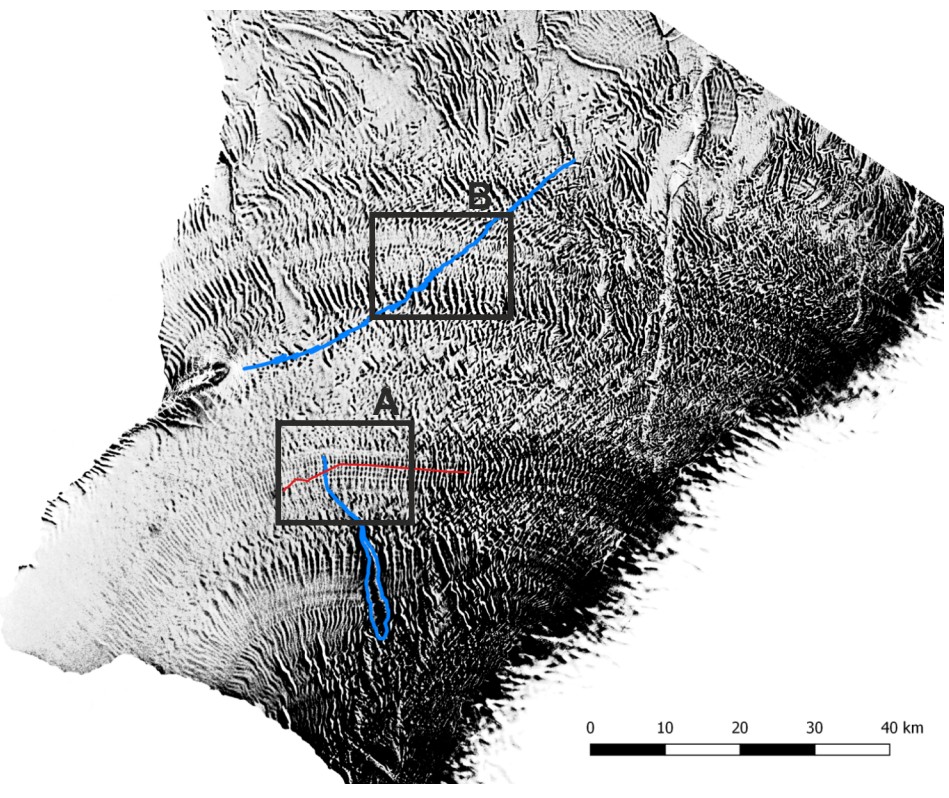

Figure 11: Map showing radar backscatter proxy for the presence of meteoric ice. Over most of the ice shelf, black indicates the presence of meteoric ice and white areas correspond to infill by sea ice and firn. Boxes indicate the location of detailed views in Figures 12 and 13. Red line is the location of the profile in Fig. 10.

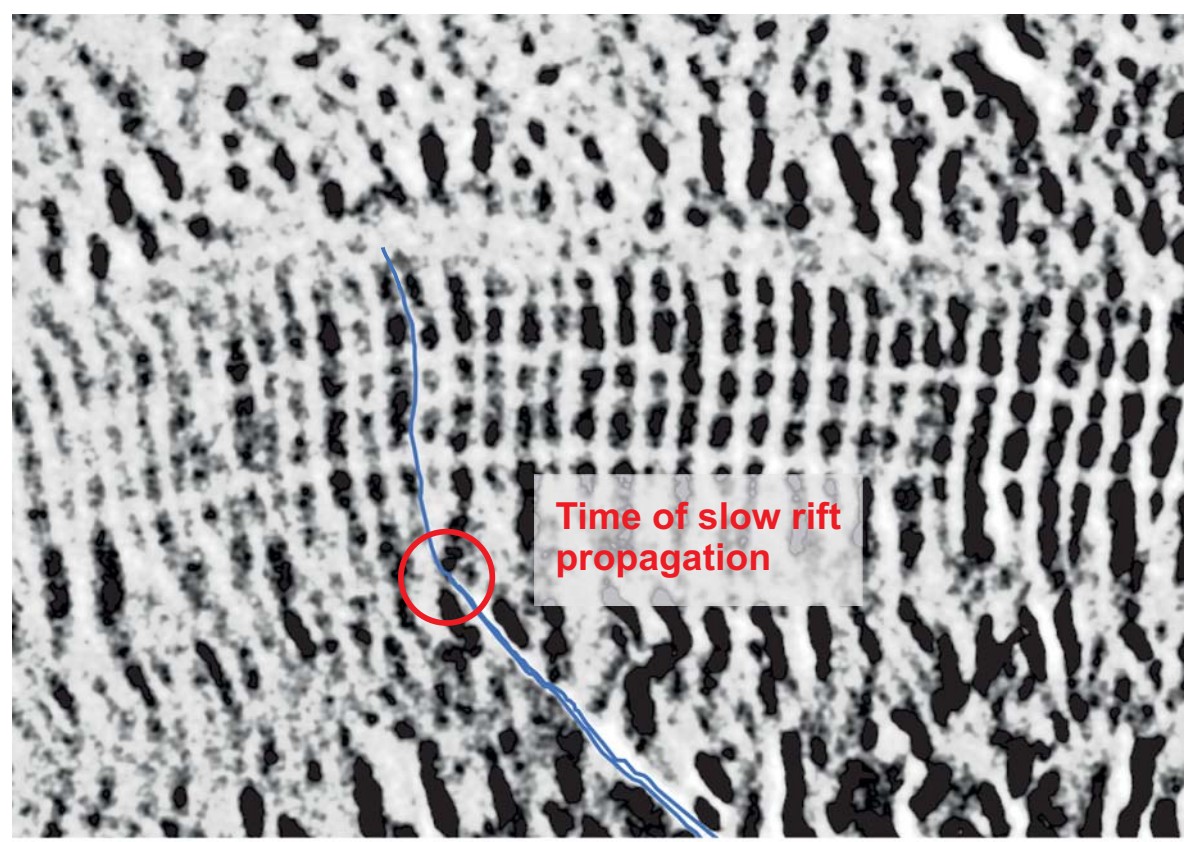

Figure 12: Detailed view of the tip of the Chasm 1 Crack (blue curve). The background image distinguishes areas with meteoric ice (dark) from areas with sea-ice overlain by firn. When the crack went through one of the blocks of meteoric ice rather than around it (red circle), the rate of propagation slowed.

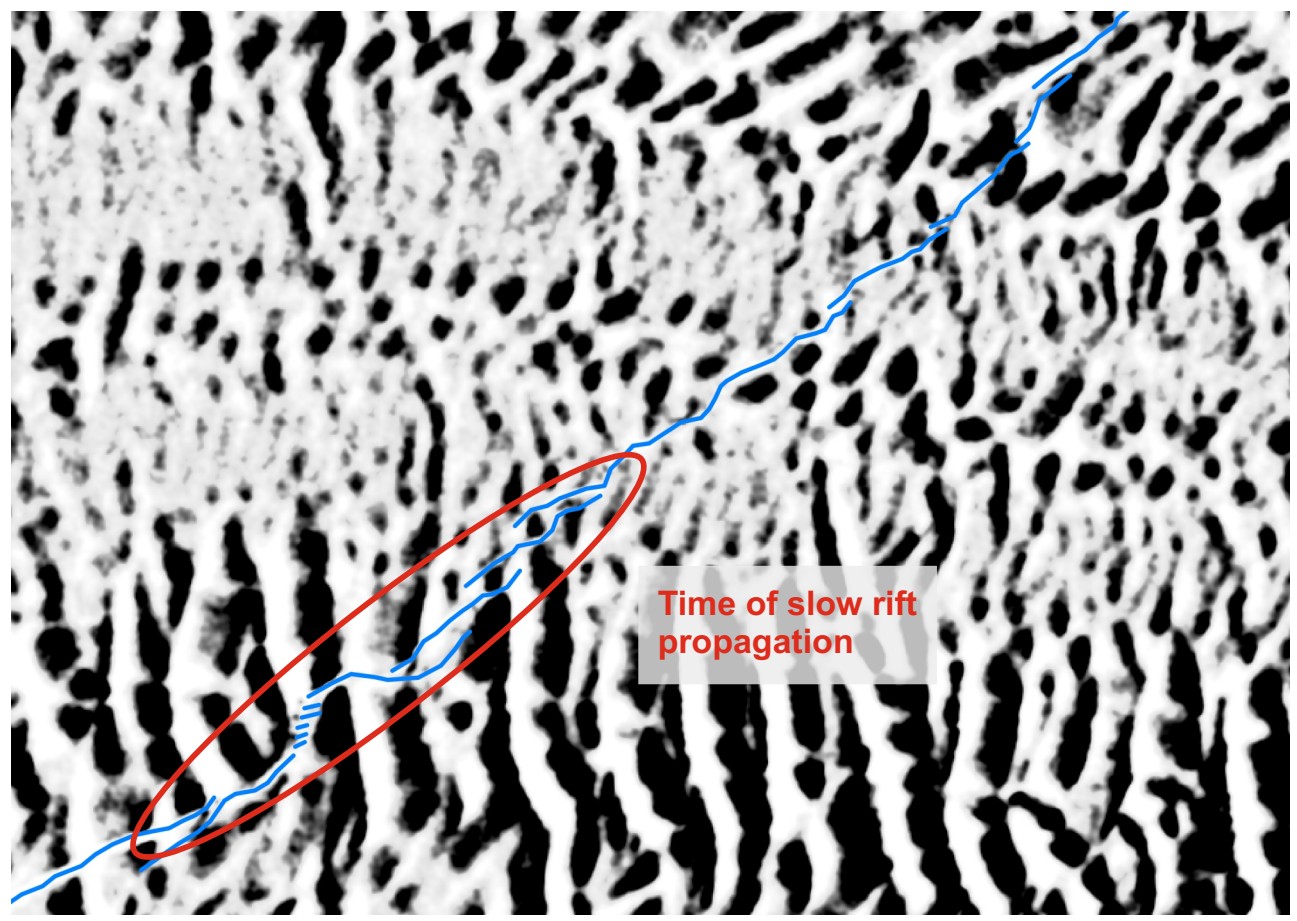

Figure 13: Detailed view of part of the Halloween Crack (blue curve). Crack propagation slowed when the crack crossed one of the 'railway tracks' at a high angle to the meteoric ice blocks. Rift extension speed then increased in the ice mélange beyond.