# Peer review of "The internal structure of the Brunt Ice Shelf from ice-penetrating radar analysis and implications for ice shelf fracture"

_The Cryosphere, 2018_

## Short Comment (SC1) · 26 Apr 2018

King et al. document the highly variable nature of the Brunt Ice Shelf and either document or hypothesize source origins and mechanisms for the various components. This worked is framed in the context of the role of ice shelf heterogeneity in controlling rift propagation. The observations are particularly unique and will make a valuable contribution to the literature on the topic. Below, I highlight a number of issues that I feel would strengthen the manuscript.

Pg 1, ln 15 – This is a unique observation (compared to rift behavior on other ice shelves) and likely reflects the significant differences in ice thickness between the me-

teoric blocks and ice melange, correct? On other ice shelves (e.g., Larsen, Amery), rift propagation is rapid through meteoric ice and slow through suture zones give their different material properties/fracture toughnesses, as you subsequently describe in the introduction. It might be worth making this point of differing behavior more explicit in the main text.

Pg 1, Ln 22: upstream dynamics of the ice sheet. This could be modified to include the "upstream and far reaching impacts of such changes", and include a citation of Reese et al., 2018.

Pg 1, Ln 23-24: Previous work by Hulbe et al. (2010) on the Ross Ice Shelf, Walker et al. (2015) on the Amery Ice Shelf, and McGrath et al. (2014) and Borstad et al. (2017) on the Larsen C ice shelf have also suggested and documented this association in detail. A more comprehensive review of the relevant literature is appropriate here since this forms the overriding context for the manuscript.

Pg 1, Ln 27: What is meant by "the suture zones can deform more rapidly without fracturing"? In previous work, authors have noted that rifts are frequently arrested by suture zones, suggesting that these flowbands have a higher fracture toughness than the neighboring meteoric ice. Can you clarify the connection between the more rapid deformation and the observations of rift arrest?

Pg 1, 26, 28:29: Is there a suitable citation for the viscosity of marine ice? In my opinion, it is overly simplistic to attribute the observations of rift arrest solely to the likely unique (but largely unmeasured) material properties of marine ice. As your findings suggest, ice mélanges and suture zones are highly heterogenous with numerous sources and numerous processes at play, and at present, I don't think we really know which of these unique characteristics (temperature, water content, altered crystal fabric, existing flaws/fractures) are responsible for arresting rift propagation. As such, it seems appropriate to acknowledge this complexity, which your findings corroborate.

Pg 2, Ln 11: What is the scale of the "large blocks"?

Pg 2, Ln 11: What is meant by the inner part of the ice shelf? Can this be labeled/marked on Figure 2?

Pg 2, Ln 22: Replace "the early system" with "previous efforts."

Pg 3, Ln 5: Given the significant topographic relief and ice velocities, how were the various DEM images from the two year interval aligned? What number of DEMs were included?

Pg 3, Ln 24 (and Figure 3): What is the source for the ice velocities?

Pg 3, Ln 25: The GPR observations are occurring over a highly heterogenous ice shelf, with likely significant spatial and vertical variations in ice density. In this context, can you describe your radar velocity derivation? Is this assumed constant across the ice shelf? If so, what uncertainties does this introduce to the analysis?

Pg 4, Ln 5:25: What are the uncertainties on these depth estimates given radar velocity uncertainties?

Pg 7, Ln 5: What is the actual statistical correlation between backscatter amplitude and the radar derived topography? Although the authors are only noting "its existence," it would be worth discussing C-band depth penetration in more detail.

Figure 6: Add scale bar.

Please include Figure 13.

References

Borstad et al., 2017, Fracture propoagtion and stability of ice shelves governed by ice shelf hetereogeneity, Geophysical Research Letters, doi:10.1002/2017GL072648.

Hulbe et al., 2010, Propagation of long fractures in the Ronne Ice Shelf, Antarctica investigated using a numerical model of fracture propagation, Journal of Glaciology, 56, 97, 459-472.

McGrath et al., 2014, The structure and effect of suture zones in the Larsen C Ice Shelf, Antarctica, JGR ES, doi:10.1002/2013JF002935

Reese et al., 2018, The far reach of ice-shelf thinning in Antarctica, Nature Climate Change, 8, 53-57, doi: 10.1038/s41558-017-0020-x

Walker et al., 2015, Observations of interannual and spatial variability in rift propagation in the Amery Ice Shelf, Antarctica, 2002-2014, Journal of Glaciology, 61, 226, doi:10.3189/2015JoG14J151.

---

## Referee Comment (RC1) · Anonymous Referee #1 · 1 May 2018

Review: "The internal structure of the Brunt Ice Shelf from ice-penetrating radar analysis and implications for ice shelf fracture" by King et al.

The authors present samples of GPR data from studies on the Brunt ice shelf in the vicinity of Halley Station, giving insights into the complicated composition of the ice shelf, and interpreting its role for fracture propagation.The paper is very well written and neatly structured, and it is a valuable contribution to research into ice shelf stability. It presents a combination of data from lots of different sources, but unfortunately only shows very little data of the long radar profiles. I have some comments and suggestions for the authors which I think should be considered before publication:

[Figure]

The authors present three short sections of GPR data, of which two are part of long profiles along flow direction of the ice shelf. It is a bit disappointing that only so little of the data is shown. Especially as the positions of the shown profiles is quite far downstream. Showing a profile from further upstream could give an indication about the temporal evolution of the ice shelf interior structures. This is not discussed at all here, but would strengthen the paper. Additionally it would be interesting to discuss the accumulation of snow respectively to the time since the ice came over the grounding line. Does this give any hints about how much sea-water infiltrated ice there is? What about the hydrostatic equilibrium? As there is a very good surface elevation model this could be easily discussed and shown?

How is the flow velocity varying along the flow line of the radar profiles? Looking at flow velocity maps of the Brunt it seems to me that that velocity is not necessarily steadily increasing along flow. Is it possible that the bending of the layers in the firn, and also the bending of the brine infiltration layer is due to shortening along flow? Is the wavelength of the surface undulations changing along flow?

Looking at Figure 2, there seems to be a rather sharp transition between the first part of the profile, from which no radar data is shown, where there are rather deep valleys between the "Railway sleepers", and the second part, where the height difference seems to level out. Is this due to the local variations of accumulation? Or might this be also due to a change in the flow regime? This would be interesting to discuss.

Is it possible from the radar data to determine the timing of the last brine infiltration by comparing the radar layers on top of the meteoric ice blocks and within the troughs? In figure 5 it looks like as if the bending brine infiltration layer to the right of the block is more or less the same isochrones as the first smooth layer on top of the block.

Are the secondary infiltration events limited to the ice between the "Railway Tracks"? This would be interesting to discuss.

The overall structure of the manuscript is very clear, but I do not understand why the

authors present the sentinel data in the last part of the discussion instead of in the observation section.

As there are a lot of figures, I am not sure whether figure 8 is really necessary. The point could be made by referring to figure 2.

The labels of the lower panels in figure 2 are hardly readable, maybe this should be changed.

---

## Referee Comment (RC2) · A. Booth (Referee) · 29 Jun 2018

I enjoyed reading this paper, and getting an insight into the internal structure of a topical ice shelf. I suspect that most of these comments should be very straightforward for the reviewers to address; my main concern is that the discussion seems a little brief, and there might be some room to consider quantifying some of the properties considered in the Conclusions.

Abstract - seems a little qualitative. Can we add a few numbers in? The ice thickness, for example? Quantify specifically what is meant by thicker and thinner bands of ice?

P2 Line 25 – possibly a little bit of a pedantic request, which potentially can't even be better-specified anyway... But might be worth adding the actual date of Halloween just to formalise the onset of rifting?

P2 Line 25 – does the heterogeneity of Brunt Ice Shelf actually make it an ideal place to study rift propagation? Presumably, you'd also want to characterise and compare it through homogeneous ice too - perhaps Brunt is actually too heterogeneous to be useful? I don't think this invalidates the study by any stretch, I just think that a more measured description could be better.

P3 Line 26 – give the manufacturer "Sensors&Software" too, and the model name is stylised PulseEKKO PRO. Also, when were the radar data acquired?

P3 Line 26 – clarify what is continuous about the acquisition mode - e.g., "towed behind a snow mobile, with traces recording continuously." Also, at what speed was the snow mobile towed, since this ultimately dictates the spatial sampling interval.

P3 Lines 1 and 2 – filter parameters required.

Section 4.1. I appreciate that Line 04, Line 05, and Line 62 are so-called because this is the naming convention given by the PulseEKKO, but for the purposes of this manuscript should this be simplified just to (e.g.) Profiles 1, 2 and 3?

What radar velocity was used to depth convert and migrate? Did you take any account of a firn gradient, or was it an average velocity?

P4 L15: Might be worth simply explaining the term 'multiple'?

P4 L30-31: Seems like a slightly irrelevant point? Almost the style of comment one would write to address a reviewer's question!

P5 L10: I agree that the horizontal reflection is likely a brine infiltration front but there are two other compelling observations that support this interpretation. First, the reflection appears in places to cross-cut the firn stratigraphy (where you don't think the

impermeable barrier is present), implying that the reflection represents a hydrological rather than stratigraphic discontinuity. Secondly, the radar signal is vastly attenuated immediately below it, suggesting a transition in an electrically conductive regime. Might be worth dropping this evidence in, in support?

P7 L30: A point of discussion rather than a recommendation here, but is the behaviour of the Halloween Crack rather contradictory to the observed propagation of the Larsen C rift? Rift propagation seemed to be slowed down when propagating through suture zones of marine ice, and accelerated through regions with more homogeneous meteoric ice. For Brunt, you're suggesting that the crack preferentially seeks to propagate through the mélange... Potentially worth a comment on this – it might add another comparative dimension to what (at the moment) reads like a bit of thin Discussion?

P8 – Conclusions. There feels like some speculation introduced here, and some of this might be better placed in the discussion. Can you comment on the absolute or relative temperature, strength and density of the ice in question?

Figures: All of generally high quality, but the captions seem over-long. They contain useful information, although some is already contained in the main text. Cut out the repetition, and shorten their overall length?

Figure 2: Axis labels in elevation profiles will likely be too small to be readable.

Figure 4: You include a Vertical Exaggeration annotation in Figure 5, so why not Figure 4 as well?

Figure 10 (and P7 Line 5): It's difficult to judge the correlation between the data in these figures; there are obviously a lot of lumps and bumps, but when presented like this it's difficult to appreciate their alignment and how well a big elevation bump corresponds to a high backscatter response. Can you derive a correlation coefficient or (possibly even better, as a third panel in this figure) cross-plot the elevation and the backscatter?

[Figure]

---

## Author Comment (AC1) · 26 Jul 2018

*Review: "The internal structure of the Brunt Ice Shelf from ice-penetrating radar analysis and implications for ice shelf fracture" by King et al.*

*The authors present samples of GPR data from studies on the Brunt ice shelf in the vicinity of Halley Station, giving insights into the complicated composition of the ice shelf, and interpreting its role for fracture propagation. The paper is very well written and neatly structured, and it is a valuable contribution to research into ice shelf stability. It presents a combination of data from lots of different sources, but unfortunately only shows very little data of the long radar profiles. I have some comments and suggestions for the authors which I think should be considered before publication:*

*The authors present three short sections of GPR data, of which two are part of long profiles along flow direction of the ice shelf. It is a bit disappointing that only so little of the data is shown. Especially as the positions of the shown profiles is quite far downstream. Showing a profile from further upstream could give an indication about the temporal evolution of the ice shelf interior structures. This is not discussed at all here, but would strengthen the paper. Additionally it would be interesting to discuss the accumulation of snow respectively to the time since the ice came over the grounding line. Does this give any hints about how much sea-water infiltrated ice there is? What about the hydrostatic equilibrium? As there is a very good surface elevation model this could be easily discussed and shown?*

**The white lines in Fig. 2 do not represent the radar profiles, they indicate the location of the surface elevation profiles, which are derived from the WorldView DEM. The caption has been changed to make this clear. The elevation profiles show that the topography becomes difficult to traverse in the region within 20 km of the grounding line, hence we did not record radar profiles in this area. The reviewer raises interesting and pertinent questions about the evolution of structures and accumulation along-flow which a future targeted study could address. The radar profiles described here were acquired as part of an operational survey to relocate Halley Station, which limited the area of investigation.**

*How is the flow velocity varying along the flow line of the radar profiles? Looking at flow velocity maps of the Brunt it seems to me that that velocity is not necessarily steadily increasing along flow. Is it possible that the bending of the layers in the firn, and also the bending of the brine infiltration layer is due to shortening along flow? Is the wavelength of the surface undulations changing along flow?*

**The reviewer is correct, the present flow speed does show variation from grounding line to ice front, there has also been historic variations in the flow speed of the entire ice shelf (Gudmundssen et al 2017). Therefore horizontal shortening may have taken place. New text added Page 5, lines 28&29, Page 7 Lines 5&6.**

*Looking at Figure 2, there seems to be a rather sharp transition between the first part of the profile, from which no radar data is shown, where there are rather deep valleys between the "Railway sleepers", and the second part, where the height difference seems to level out. Is this due to the local variations of accumulation? Or might this be also due to a change in the flow regime? This would be interesting to discuss.*

**In the absence of ground truth information from the inner part of the ice shelf, we do not feel we have sufficient information to discuss this. The flow speed increase in about 1971 is likely a**

**significant factor but some dated accumulation profiles either side of the change would be needed to prove this.**

*Is it possible from the radar data to determine the timing of the last brine infiltration by comparing the radar layers on top of the meteoric ice blocks and within the troughs? In figure 5 it looks like as if the bending brine infiltration layer to the right of the block is more or less the same isochrones as the first smooth layer on top of the block.*

**We do not think that dating brine infiltration can be done by relating the brine reflection to isochrones in the firn. There are many places where the brine reflection cross-cuts the firn isochrones (most clear in Fig. 7). The cartoon in Fig 8 seeks to demonstrate that brine infiltration from below can occur as the firn is pushed below sea level by the weight of accumulation above, so that there is no time correspondence between firn deposition and brine infiltration.**

*Are the secondary infiltration events limited to the ice between the "Railway Tracks"? This would be interesting to discuss.*

**If the reviewer means by secondary infiltration the kind labelled in Fig. 7 as 'Recent brine infiltration', then this has only been observed immediately adjacent to the presently-active rift called Chasm 1.**

*The overall structure of the manuscript is very clear, but I do not understand why the authors present the sentinel data in the last part of the discussion instead of in the observation section.*

**We feel that the Sentinel-1 data can only be understood after the basics of the internal structure have been established.  It is an aid to interpretation rather than a primary observation. Its particular application is in establishing how fractures interact with the internal structure and we feel that is more of a discussion of what the structure results mean.**

*As there are a lot of figures, I am not sure whether figure 8 is really necessary. The point could be made by referring to figure 2.*

**We disagree because Fig 8 summarises the production of the two types of structure in a readily accessible and memorable form.**

*The labels of the lower panels in figure 2 are hardly readable, maybe this should be changed.*

**Agreed, figure changed.**
*King et al. document the highly variable nature of the Brunt Ice Shelf and either document or hypothesize source origins and mechanisms for the various components. This worked is framed in the context of the role of ice shelf heterogeneity in controlling rift propagation. The observations are particularly unique and will make a valuable contribution to the literature on the topic. Below, I highlight a number of issues that I feel would strengthen the manuscript.*

*Pg 1, ln 15 – This is a unique observation (compared to rift behavior on other ice shelves) and likely reflects the significant differences in ice thickness between the meteoric blocks and ice melange, correct? On other ice shelves (e.g., Larsen, Amery), rift propagation is rapid through meteoric ice and slow through suture zones give their different material properties/fracture toughnesses, as you subsequently describe in the introduction. It might be worth making this point of differing behavior more explicit in the main text.*

**The contrast with other ice shelves has been added to the Abstract as well as sections 1, 5, 6 and 7**

*Pg 1, Ln 22: upstream dynamics of the ice sheet. This could be modified to include the "upstream and far reaching impacts of such changes", and include a citation of Reese et al., 2018.*

**Done**

*Pg 1, Ln 23-24: Previous work by Hulbe et al. (2010) on the Ross Ice Shelf, Walker et al. (2015) on the Amery Ice Shelf, and McGrath et al. (2014) and Borstad et al. (2017) on the Larsen C ice shelf have also suggested and documented this association in detail. A more comprehensive review of the relevant literature is appropriate here since this forms the overriding context for the manuscript.*

**Done**

*Pg 1, Ln 27: What is meant by "the suture zones can deform more rapidly without fracturing"? In previous work, authors have noted that rifts are frequently arrested by suture zones, suggesting that these flowbands have a higher fracture toughness than the neighboring meteoric ice. Can you clarify the connection between the more rapid deformation and the observations of rift arrest?*

**Phrase removed and replaced with reference to fracture toughness as part of improved literature review.**

*Pg 1, 26, 28:29: Is there a suitable citation for the viscosity of marine ice? In my opinion, it is overly simplistic to attribute the observations of rift arrest solely to the likely unique (but largely unmeasured) material properties of marine ice. As your findings suggest, ice mélanges and suture zones are highly heterogenous with numerous sources and numerous processes at play, and at present, I don't think we really know which of these unique characteristics (temperature, water content, altered crystal fabric, existing flaws/fractures) are responsible for arresting rift propagation. As such, it seems appropriate to acknowledge this complexity, which your findings corroborate.*

**Done, see improved literature review.**

*Pg 2, Ln 11: What is the scale of the "large blocks"?*

**Figures added to provide the scale. Pg 2 ln 22**.

*Pg 2, Ln 11: What is meant by the inner part of the ice shelf? Can this be labeled/marked on Figure 2?*

**Figures added to text to define 'inner part'. Pg2, ln 23.**

*Pg 2, Ln 22: Replace "the early system" with "previous efforts."*

**Done.**

*Pg 3, Ln 5: Given the significant topographic relief and ice velocities, how were the various DEM images from the two year interval aligned? What number of DEMs were included?*

**Text added, Section 2.1**

*Pg 3, Ln 24 (and Figure 3): What is the source for the ice velocities?*

**Added reference to MEASURES velocity product (Rignot et al)**

*Pg 3, Ln 25: The GPR observations are occurring over a highly heterogenous ice shelf, with likely significant spatial and vertical variations in ice density. In this context, can you describe your radar velocity derivation? Is this assumed constant across the ice shelf? If so, what uncertainties does this introduce to the analysis?*

**Text added to section 3. We use a uniform wave speed for both migration and depth conversion because we do not have CMP data to generate a wave speed profile. We are not attempting any quantative analysis in this paper**

*Pg 4, Ln 5:25: What are the uncertainties on these depth estimates given radar velocity uncertainties?*

**Text added in Section 3**

*Pg 7, Ln 5: What is the actual statistical correlation between backscatter amplitude and the radar derived topography? Although the authors are only noting "its existence," it would be worth discussing C-band depth penetration in more detail.*

**Correlation is r = 0.607, p = 0. We do not agree that a discussion of C-band penetration will enhance this paper but we do think that the radar dataset from the Brunt could be used for a quantitative analysis of the Sentinel-1 backscatter data.**

*Figure 6: Add scale bar.*

**Added axes labels to show grid dimensions.**

*Please include Figure 13.*

**Done**

*A. Booth (Referee) a.d.booth@leeds.ac.uk*

*I enjoyed reading this paper, and getting an insight into the internal structure of a topical ice shelf. I suspect that most of these comments should be very straightforward for the reviewers to address; my main concern is that the discussion seems a little brief, and there might be some room to consider quantifying some of the properties considered in the Conclusions.*

*Abstract - seems a little qualitative. Can we add a few numbers in? The ice thickness, for example? Quantify specifically what is meant by thicker and thinner bands of ice?*

**Values added.**

*Line 25 – possibly a little bit of a pedantic request, which potentially can't even be better-specified anyway. . . But might be worth adding the actual date of Halloween just to formalise the onset of rifting?*

**Done (yes, a little bit)**

*P2 Line 25 – does the heterogeneity of Brunt Ice Shelf actually make it an ideal place to study rift propagation? Presumably, you'd also want to characterise and compare it through homogeneous ice too - perhaps Brunt is actually too heterogeneous to be useful? I don't think this invalidates the study by any stretch, I just think that a more measured description could be better.*

**'ideal' replaced by 'unique' to recognise that the Brunt may be an end-member ice shelf.**

*P3 Line 26 – give the manufacturer "Sensors&Software" too, and the model name is stylised PulseEKKO PRO. Also, when were the radar data acquired?*

**Done, Section 3.**

*P3 Line 26 – clarify what is continuous about the acquisition mode - e.g., "towed behind a snow mobile, with traces recording continuously." Also, at what speed was the snow mobile towed, since this ultimately dictates the spatial sampling interval.*

**Done, Section 3.**

*P3 Lines 1 and 2 – filter parameters required.*

**Done, Section 3.**

*Section 4.1. I appreciate that Line 04, Line 05, and Line 62 are so-called because this is the naming convention given by the PulseEKKO, but for the purposes of this manuscript should this be simplified just to (e.g.) Profiles 1, 2 and 3?*

**Disagree. A link should be maintained between data that is published and any archive of those data. Renumbering profiles in papers unnecessarily breaks that link with no real benefit to the reader.**

*What radar velocity was used to depth convert and migrate? Did you take any account of a firn gradient, or was it an average velocity?*

**We used a single wave speed value (0.168 m/ns) for both migration and depth conversion in the absence of any data. Text added to Section 3.**

*P4 L15: Might be worth simply explaining the term 'multiple'?*

**Done**

*P4 L30-31: Seems like a slightly irrelevant point? Almost the style of comment one would write to address a reviewer's question!*

**Left in. This explanation is in the same class as explaining the term 'multiple' which was the reviewer's last recommendation – it helps none-geophysicists to understand concepts and terminology that are common currency within the discipline.**

*P5 L10: I agree that the horizontal reflection is likely a brine infiltration front but there are two other compelling observations that support this interpretation. First, the reflection appears in places to cross-cut the firn stratigraphy (where you don't think the impermeable barrier is present), implying that the reflection represents a hydrological rather than stratigraphic discontinuity. Secondly, the radar signal is vastly attenuated immediately below it, suggesting a transition in an electrically conductive regime. Might be worth dropping this evidence in, in support?*

**Done**

*P7 L30: A point of discussion rather than a recommendation here, but is the behaviour of the Halloween Crack rather contradictory to the observed propagation of the Larsen C rift? Rift propagation seemed to be slowed down when propagating through suture zones of marine ice, and accelerated through regions with more homogeneous meteoric ice. For Brunt, you're suggesting that the crack preferentially seeks to propagate through the mélange. . . Potentially worth a comment on this – it might add another comparative dimension to what (at the moment) reads like a bit of thin Discussion?*

**Text on this contrasting behaviour now included in the Abstract and Sections 1, 5, 6 and 7**

*P8 – Conclusions. There feels like some speculation introduced here, and some of this might be better placed in the discussion. Can you comment on the absolute or relative temperature, strength and density of the ice in question?*

**More speculative material moved to discussion. Conclusions re-written.**

*Figures: All of generally high quality, but the captions seem over-long. They contain useful information, although some is already contained in the main text. Cut out the repetition, and shorten their overall length?*

**Caption shortening done.**

*Figure 2: Axis labels in elevation profiles will likely be too small to be readable.*

**Fixed**

*Figure 4: You include a Vertical Exaggeration annotation in Figure 5, so why not Figure 4 as well?*

**Done**

*Figure 10 (and P7 Line 5): It's difficult to judge the correlation between the data in these figures; there are obviously a lot of lumps and bumps, but when presented like this it's difficult to appreciate their alignment and how well a big elevation bump corresponds to a high backscatter*

*response. Can you derive a correlation coefficient or (possibly even better, as a third panel in this figure) cross-plot the elevation and the backscatter?*

**Text added to give correlation values.**

---

## Author Comment (AC2)

[revised manuscript text omitted]

Time of slow rift propagation